# Ethics and legal requirements for data linkage in 14 European countries for children with congenital anomalies

Hugh Claridge [ID],[1] Joachim Tan,[2] Maria Loane [ID],[3] Ester Garne [ID],[4] Ingeborg Barisic,[5] Clara Cavero-Carbonell [ID],[6] Carlos Dias [ID],[7] Miriam Gatt,[8] Susan Jordan [ID],[9] Babak Khoshnood [ID],[10] Sonja Kiuru-Kuhlefelt,[11] Kari Klungsoyr [ID],[12,13] Olatz Mokoroa Carollo,[14] Vera Nelen,[15] Amanda J Neville,[16] Anna Pierini,[17] Hanitra Randrianaivo,[18] Anke Rissmann [ID],[19] David Tucker,[20] Hermien de Walle,[21] Wladimir Wertelecki,[22] Joan K Morris [ID] [2]

For numbered affiliations see end of article.

**Correspondence to**
Joan K Morris;
jmorris@sgul.ac.uk

## ABSTRACT

**Introduction** Linking healthcare data sets can create valuable resources for research, particularly when investigating rare exposures or outcomes. However, across Europe, the permissions processes required to access data can be complex. This paper documents the processes required by the EUROlinkCAT study investigators to research the health and survival of children with congenital anomalies in Europe.

**Methods** Eighteen congenital anomaly registries in 14 countries provided information on all the permissions required to perform surveillance of congenital anomalies and to link their data on live births with available vital statistics and healthcare databases for research. Small number restrictions imposed by data providers were also documented.

**Results** The permissions requirements varied substantially, with certain registries able to conduct congenital anomaly surveillance as part of national or regional healthcare provision, while others were required to obtain ethics approvals or informed consent. Data linkage and analysis for research purposes added additional layers of complexity for registries, with some required to obtain several permissions, including ethics approvals to link the data. Restrictions relating to small numbers often resulted in a registry's data on specific congenital anomalies being unusable.

**Conclusion** The permissions required to obtain and link data on children with congenital anomalies varied greatly across Europe. The variation and complexity present a significant obstacle to the use of such data, especially in large data linkage projects. Furthermore, small number restrictions severely limited the research that could be performed for children with specific rare congenital anomalies.

## INTRODUCTION

The EUROlinkCAT study aimed to support 22 congenital anomaly registries in 14 countries to link their data on live births with congenital anomalies to other regional or national electronic healthcare databases, to investigate the survival and morbidity of these children up to 10 years of age. This article briefly summarises the different legal, ethical, governance and other permissions that the registries had to obtain to perform this work.

## METHODS

Each congenital anomaly registry was required to submit to the EUROlinkCAT Coordinator (St George's, University of London), the permissions for the registry to fulfil its surveillance remit and to send data to EUROCAT (the European network of population-based congenital anomaly registries). An umbrella protocol was then supplied to each registry, providing sufficient information for them to apply to the relevant authorities to enable them to link their data on liveborn children with congenital anomalies to national and regional electronic databases containing information on the survival and morbidity of these children. The registries were responsible for obtaining the necessary permissions, and once obtained these were submitted to the EUROlinkCAT Coordinator. This article summarises all the permissions obtained. As the permissions needed for surveillance differed from those needed for the subsequent research, details regarding the two sets of permissions are reported separately.

The English data on congenital anomalies included in the EUROlinkCAT project were provided by five regional congenital anomaly registries. However, in April 2015, the regional registries in England were all combined into the National Congenital Anomaly and Rare Disease Registration Service (NCARDRS). As the regional registries no longer exist, the permissions that would be required to obtain

data from NCARDRS are reported here instead. This is why information is available for 18 registries in total, rather than the original 22.

Information from 18 registries in 14 countries are presented, relating to the timeframe of the EUROlinkCAT project from 2017 to 2022—it is therefore possible that some permission requirements have since changed.

The answers from the registries are categorised in table 1 under the following headings:

1. Permissions required for registry to perform surveillance of congenital anomalies.
   1. Permission granted as part of regional/national healthcare provision.
      1. Yes: National or regional legislation exists that enables the registry to collect data for surveillance purposes.
      2. No: National or regional legislation does not exist to enable the registry to collect data for surveillance purposes.
   2. Permission from hosting institution.
      1. Yes: The registry needs to obtain permission to collect data for surveillance.
      2. No: No additional permission is needed as national/regional legislation is sufficient.
   3. Ethics permission.
      1. Yes: The registry needs to obtain ethics permission to collect data for surveillance.
      2. No: The registry does not need to obtain ethics permission to collect data for surveillance.
   4. Informed consent needed.
      1. Yes: The registry needs to obtain informed consent from the parents to register the congenital anomaly.
      2. No: The registry does not need to obtain informed consent from the parents to register the congenital anomaly. However, the parents can choose to have their details removed by contacting the relevant organisation.
2. Permissions required for registry to link data and analyse for research purposes.
   1. Ethics.
      1. Yes: Ethics permission is needed for research projects to link and analyse the linked data.
      2. No: Ethics permission is not needed for research projects using the linked data.
   2. Other permissions needed.
      1. Yes: Other permissions, not including ethics, such as approval by the data provider regarding the safety of the data storage, permission to share the data with other named parties or permission to release the analytic results for publication are required.
      2. No: No additional permissions are needed for research using the linked data.
   3. Small number restrictions.
      1. Yes-Pub: Small number restrictions apply to any publication of the data, but prepublication results provided to named researchers may include small numbers.
      2. Yes-Int: Small number restrictions apply to all use of data including prepublication results.
      3. No: No small number restrictions need to be applied.

## RESULTS

### Permissions required for surveillance of congenital anomalies

The majority (16/18) of registries have permission to perform surveillance of congenital anomalies as part of their national or regional healthcare provision. Of these 16, three are required to obtain ethics approval to perform this surveillance and one is required to obtain informed consent from the parents/guardians of the children. Of the two registries that did not have permission under their national or regional healthcare provision, one needed permission to collect the data from their host institution and one needed both ethics permission and informed consent.

### Permissions required for data linkage and analysis for research purposes

Of the six registries who only linked their data to mortality data, five required specific ethics permission to do so, and three of these also required other additional permissions. Of the 12 registries who linked their data to electronic healthcare databases, eight required specific ethics permission to do so and four of these required other additional permissions. The four registries who did not require specific ethics permission to link their data to electronic healthcare databases, did however require other additional permissions, such as demonstrating to the data providers that the data would be stored safely, that the project was of sufficient scientific merit and providing other assurances arising from the European Union's General Data Protection Regulation (GDPR) 2016/679 requiring that the data be used in a transparent manner.

Once the data had been linked to other databases, the authorities providing these data often imposed strict restrictions on the publication of results when small numbers of children were involved. This was to ensure that children with rare congenital anomalies and their families could not be identified. Some authorities did allow aggregate results from small numbers of individuals to be distributed among named researchers, with the requirement that these results could only be published when combined with data from other registries. Permitting the internal distribution among specified researchers was extremely helpful, as it enabled meta-analyses over several European countries to be performed for groups of rare congenital anomalies and combined results to be published. Authorities suppressing the provision of small numbers of cases for analysis (eg, through rounding counts to the nearest 5) often resulted in a registry's data being excluded from the meta-analyses due to

**Table 1** The permissions required for congenital anomaly registries to perform surveillance and research

| Full name of CA registry (geographical region) | Permissions required for registry to perform surveillance of congenital anomalies | | | Permissions required for registry to link data and analyse for research purposes | | | | Notes |
|---|---|---|---|---|---|---|---|---|
| | Permission granted as part of regional/national healthcare provision | Permission from hosting institution | Ethics permission | Informed consent | Ethics permission | Other permissions | Small number restrictions | |
| Antwerp Congenital Anomaly Registry (Belgium: Antwerp) | Yes | No | No | No | Yes | Yes | Yes-Int | Considered linkage to mortality data only |
| Zagreb Congenital Anomaly Registry (Croatia: Zagreb) | No | Yes | No | No | Yes | No | No | |
| Odense (Denmark: Funen) | Yes | Yes | No | No | No | Yes | Yes-Pub | |
| Finnish Register of Congenital Malformations (Finland: all) | Yes | No | No | No | Yes | No | No | |
| Registre des malformations congénitales de l'île de la Réunion (France: Île de la Réunion) | Yes | No | Yes | No | Yes | Yes | No | |
| Registre des malformations congénitales de Paris (France: Paris) | Yes | No | Yes | No | Yes | Yes | No | Considered linkage to mortality data only |
| Saxony-Anhalt Congenital Anomaly Registry (Germany: Saxony-Anhalt) | No | No | Yes | Yes | Yes | No | No | Manual linkage only to mortality data due to Bundesstatistikgesetz $21 (Prohibition of re-identification) |
| IMER Registry (Italy: Emilia Romagna) | Yes | No | No | No | No | Yes | No | |
| Tuscany Congenital Anomaly Registry (Italy: Tuscany) | Yes | No | No | No | No | Yes | No | |
| Malta Congenital Anomaly Registry (Malta: all) | Yes | No | No | No | Yes | No | No | Considered linkage to mortality data only |
| EUROCAT Registry - Northern Netherlands (Netherlands: Northern) | Yes | No | No | Yes | Yes | Yes | Yes-Int | Other permissions: parents are given the option to opt out of EUROlinkCAT |
| Medical Birth Registry of Norway (Norway: All) | Yes | No | No | No | Yes | Yes | No | Considered linkage to mortality data only |
| RENAC: Registo Nacional de Anomalias Congénitas (Portugal: South) | Yes | Yes | No | No | Yes | Yes | No | |
| Basque Congenital Anomaly Registry (Spain: Basque Country) | Yes | No | No | No | Yes | No | No | |
| Valencian Region Congenital Anomalies Registry (Spain: Valencian Region) | Yes | No | No | No | Yes | No | No | |

Continued

**Table 1** Continued

| Full name of CA registry (geographical region) | Permissions required for registry to perform surveillance of congenital anomalies | | | | Permissions required for registry to link data and analyse for research purposes | | | Notes |
| | Permission granted as part of regional/national healthcare provision | Permission from hosting institution | Ethics permission | Informed consent | Ethics permission | Other permissions | Small number restrictions | |
| --- | --- | --- | --- | --- | --- | --- | --- | --- |
| National Congenital Anomaly and Rare Disease Registration Service (NCARDRS) (UK: England) | Yes | No | No | No | Yes | Yes | Yes-Pub | NCARDRS became part of NHS Digital in October 2021 and access conditions are currently under review |
| Congenital Anomaly Register and Information Service (CARIS) linked with SAIL databank (UK: Wales) | Yes | No | Yes | No | No | Yes | Yes-Pub | |
| OMNI-Net Ukraine Birth Defects Programme (Ukraine: West) | Yes | No | No | No | No | No | No | Considered linkage to mortality data only |

CA, Congenital Anomaly.

the rounding having a significant effect on the overall estimates.

The length of time required to obtain the necessary permissions varied considerably by registry, with it taking over two years for several registries to obtain the permissions and thus over two years to obtain the final linked data. In contrast, some registries were granted approvals within three months.

## DISCUSSION

To the authors' knowledge, this is the first example in the literature of a multicentre, multinational health data linkage project documenting the broad range of permissions required across Europe to access and link data on children with congenital anomalies. The increased awareness that this article provides of the high variability in data providers' requirements and the timescales that the permissions-related processes operate within, will be of great assistance to funding bodies and both current and future studies in this and related fields. For multicentre studies, the widely variable waiting times make it very challenging to determine a common timeframe for the overall study to work to. Indeed, the challenges faced by EUROlinkCAT researchers are not unique, with researchers across the world facing similar issues (UK,[1] Australia[2] and USA[3]). Advanced knowledge of the challenges means staff time and thus personnel budgets can be more accurately costed over a project's timeframe; awareness of the possibility of permissions interdependencies can permit improved planning and more efficient task prioritisation; data access expiration dates can be more appropriately chosen to limit the need for amendments and reapplications, and staff can be better prepared for the sometimes demoralising wait for permissions to be granted.

The suppression of small numbers, such as preventing the release of numbers smaller than 10 even for internal analysis purposes, has a seriously detrimental impact on rare disease research. To illustrate this problem, consider Ebstein's Anomaly, which is a severe rare congenital heart defect with a mortality of around 50% in the first month of life,[4] and continued high mortality in the first five years of life.[5] The prevalence of Ebstein's Anomaly is about 4 per 100 000 births.[6] EUROlinkCAT registries have survival information on approximately 400 live births with Ebstein's Anomaly. If the numbers of children dying during the first month of life, the numbers dying after the first month of life and before their first birthday, and the numbers surviving past the age of one are required, all 22 EUROlinkCAT registries would be likely to have at least one out of the three cells below 10 and therefore none of the registries would be able to provide any aggregated data. Hence, no information about mortality during the first year of life could be obtained, despite a sample size of 400 children from a study period of 20 years and a population of 10 million births.

The rationale for small number restrictions is to prevent individuals or their families being identified. A distinction needs to be made between data released for publication and data released to named researchers for analysis. The implementation of GDPR across Europe in 2018 has ensured that all European countries have structures in place to ensure the safe use of an individual's data. These regulations should be applied in such a way as to allow all European countries to provide aggregate data to be used for pan-European analyses across countries to trusted, named researchers, even if certain categories of such aggregate data comprise fewer than 10 cases.

## CONCLUSIONS

There is a lack of consistency across Europe in the permissions required to access and link data on children with congenital anomalies. The complexity and time taken to obtain such permissions present a significant obstacle to the use of such data. In addition, the imposition of small number restrictions has a detrimental effect on the analyses that can be performed for children with rare anomalies.

### Author affiliations
[1] School of Health Sciences, Faculty of Health and Medical Sciences, University of Surrey, Guildford, UK
[2] Population Health Research Institute, St George's, University of London, London, UK
[3] Faculty of Life and Health Sciences, Ulster University, Belfast, UK
[4] Department of Paediatrics and Adolescent Medicine, Lillebaelt Hospital, University Hospital of Southern Denmark, Kolding, Denmark
[5] Children's Hospital Zagreb, Centre of Excellence for Reproductive and Regenerative Medicine, Medical School University of Zagreb, Zagreb, Croatia
[6] Rare Diseases Research Unit, Foundation for the Promotion of Health and Biomedical Research in the Valencian Region (FISABIO), Valencia, Spain
[7] Epidemiology Department, National Registry of Congenital Anomalies, National Institute of Health Doctor Ricardo Jorge (Instituto Nacional de Saúde Doutor Ricardo Jorge), Lisbon, Portugal
[8] Malta Congenital Anomalies Registry, Directorate for Health Information and Research, Pieta, Malta
[9] Faculty of Medicine, Health and Life Sciences, Swansea University, Swansea, UK
[10] Obstetrical, Perinatal and Pediatric Epidemiology Research Team (EPOPé), Center of Research in Epidemiology and Statistics (CRESS), Institut National de la Santé et de la Recherche Médicale (INSERM), INRA, Université de Paris, Paris, France
[11] Knowledge Brokers, Finnish Institute for Health and Welfare, Helsinki, Finland
[12] Department of Global Public Health and Primary Care, University of Bergen, Bergen, Norway
[13] Divison of Mental and Physical Health, Norwegian Institute of Public Health, Bergen, Norway
[14] Public Health Division of Gipuzkoa, BioDonostia Health Research Institute, San Sebastian, Spain
[15] Provincial Institute for Hygiene, Antwerp, Belgium
[16] Registro IMER, University of Ferrara, Ferrara, Emilia-Romagna, Italy
[17] Institute of Clinical Physiology, National Research Council, Pisa, Italy
[18] Register of Congenital Malformations, Centre Hospitalier Universitaire de La Réunion, Île de la Réunion, France
[19] Malformation Monitoring Centre Saxony-Anhalt, Medical Faculty, Otto-von-Guericke University, Magdeburg, Germany
[20] Public Health Wales National Health Service Trust, Cardiff, UK
[21] Department of Genetics, University Medical Center Groningen, University of Groningen, Groningen, The Netherlands
[22] OMNI-Net for Children International Charitable Fund, Rivne, Ukraine

**Contributors** JKM is the principal author and the project's scientific coordinator. HC is the senior project manager. JT is the senior statistician. ML is the data coordinator. EG is the clinical coordinator and EUROCAT registry leader. IB, CC-C, CD, MG, SJ, BK, SK-K, KK, OMC, VN, AN, AP, HR, AR, DT, HdW and WW are EUROCAT registry leaders. All authors contributed to, read and approved the final manuscript.

**Competing interests** None declared.

**Ethics approval** Not required.

**Provenance and peer review** Not commissioned; externally peer reviewed.

### ORCID iDs
Hugh Claridge http://orcid.org/0000-0001-5998-2860
Maria Loane http://orcid.org/0000-0002-1206-3637
Ester Garne http://orcid.org/0000-0003-0430-2594
Clara Cavero-Carbonell http://orcid.org/0000-0002-4858-6456
Carlos Dias http://orcid.org/0000-0002-0206-5874
Susan Jordan http://orcid.org/0000-0002-5691-2987
Babak Khoshnood http://orcid.org/0000-0002-4031-4915
Kari Klungsoyr http://orcid.org/0000-0003-2482-1690
Anke Rissmann http://orcid.org/0000-0002-9437-2790
Joan K Morris http://orcid.org/0000-0002-7164-612X

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
