## [Reviewer comments · BMJ Open]

ARTICLE DETAILS

TITLE (PROVISIONAL)	Ethics and legal requirements for data linkage in 14 European countries for children with congenital anomalies
AUTHORS	Claridge, Hugh; Tan, Joachim; Loane, Maria; Garne, Ester; Barisic, Ingeborg; Caverio- Carbonell, Clara; Dias, Carlos; Gatt, Miriam; Jordan, Susan; Khoshnood, Babak; Kiuru-Kuhlefelt, Sonja; Klungsoyr, Kari; Mokoroa Carollo, Olatz; Nelen, Vera; Neville, Amanda; Pierini, Anna; Randrianaivo, Hanitra; Rissmann, Anke; Tucker, David; de Walle, Hermien; Wertelecki, Wladimir; Morris, Joan

VERSION 1 – REVIEW

REVIEWER	Lotto, Robyn Liverpool John Moores University, School of Nursing and Allied Health
REVIEW RETURNED	28-Jan-2023

GENERAL COMMENTS	This article made for an interesting (and depressing) read. Whilst, anecdotally, issues around variations in ethical requirements are discussed around grant planning tables, I am not aware of any articles bringing these variations together in this way. In particular, the suppression of small numbers is not something I had previously considered, and a strong argument is made to remove these barriers. I would be interested to know if the authors have any suggestions on how to start addressing these issues?
---

REVIEWER	Feltbower, Richard University of Leeds
REVIEW RETURNED	17-Feb-2023

GENERAL COMMENTS	Claridge – BMJ Open 2023 The authors describe their experience in attempts to link congenital anomaly data from the EUROlinkCAT study across 14 European countries and 22 disease registries with other local or national health datasets to examine survival and morbidity in this vulnerable population. They explain that the pooling of information and data linkage approaches are the only solution to evaluating rare diseases such as congenital anomalies in a robust manner,. However, a major challenge is the difficulty in sharing rare disease data into a single repository for analysis whilst adhering to various ethical and data sharing permissions within each country. Furthermore, the vagaries of small number suppression rules which are often needed for pooled meta analyses is also carefully explained with a clear example and the point well made. In summary, the article covers a timely topic which will be of interest
---

	to clinicians and epidemiologists throughout the world wishing to share linked patient data across borders in a safe and secure way. Main comment The authors present the case in terms of permissions required to perform surveillance and research, e.g. Table 1. However, certain jurisdictions differentiate the purpose for collecting and processing personal data between research and 'non-research', the latter including disease surveillance, service evaluation or clinical audit. For example in the UK or at least England, ethical approval is required for research purposes but not for 'non-research' projects. Furthermore, if an 'opt out' consent model is adopted rather than explicit 'opt in' consent, separate information governance applications are required to cover research and 'non-research' purposes: this is the case for England but not for Scotland. I wondered therefore if the authors wanted to highlight this subtle but important distinction between research and surveillance as another example of the added complexity of processing personal data in certain countries, if space allows. Minor Comments Table 1 - column width formatting to ensure the end of words are not split onto 2nd lines.
--	---

VERSION 1 – AUTHOR RESPONSE

Thank you for your reviews of the submitted manuscript and for your suggestions and comments regarding the paper's content; we have separated out the responses below for ease of discussion.

Comments from Dr Lotto: The authors would like to thank Dr Lotto for their comments on the paper. Regarding whether we have suggestions on how to start addressing the issues presented in the paper, our most concrete recommendations as detailed in the paper relate to small number suppression and the need for a distinction to be made between data released for publication and data analysed by named researchers. Using the structures implemented through GDPR to ensure the safe use of individuals' data, we suggest that all European countries ought to be able to provide aggregate data to be used for pan-European analysis across countries to named researchers, even if certain categories of such aggregate data comprise fewer than 10 cases. With regards to the issue of the wide variation in permissions required for data linkage across Europe, we felt it important to bring these issues to the attention of the journal's readership so that a wider audience could make recommendations on how to improve this situation. Currently this barrier to pan-European research will impede the progress of research projects for years to come, and we hope that by drawing attention to it, the valuable resources that are currently spent on overcoming these impediments can in future be better spent on performing the research itself.

Comments from Prof Feltbower: The authors thank Prof Feltbower for their comments and suggested addition to the paper. The authors have therefore added the following into the manuscript in the 9th line of the Methods section to further highlight the different permissions needed for surveillance activities and research activities: 'As the permissions needed for surveillance differed from those needed for the subsequent research, details regarding the two sets of permissions are reported separately.' Thank you also Prof Feltbower for bringing the table formatting issue to our attention – we have corrected this in the submitted manuscript.

The authors would again like to thank the Reviewers Dr Lotto and Prof Feltbower for their review of the article.

VERSION 2 – REVIEW

REVIEWER	Lotto, Robyn Liverpool John Moores University, School of Nursing and Allied Health
REVIEW RETURNED	16-Mar-2023
GENERAL COMMENTS	I really enjoyed reading the paper. No comments to add.
REVIEWER	Feltbower, Richard University of Leeds
REVIEW RETURNED	09-Mar-2023
GENERAL COMMENTS	Happy with the authors' responses and changes to the manuscript following the reviewers' comments.